# Male sex identified by global COVID-19 meta-analysis as a risk factor for death and ITU admission

Hannah Peckham [1,2], Nina M. de Gruijter [1,2], Charles Raine [2], Anna Radziszewska [1,2], Coziana Ciurtin [1,2], Lucy R. Wedderburn [1,3,4], Elizabeth C. Rosser [1,2,7], Kate Webb [5,6,7✉] & Claire T. Deakin [1,3,4,7✉]

Anecdotal evidence suggests that Coronavirus disease 2019 (COVID-19), caused by the coronavirus SARS-CoV-2, exhibits differences in morbidity and mortality between sexes. Here, we present a meta-analysis of 3,111,714 reported global cases to demonstrate that, whilst there is no difference in the proportion of males and females with confirmed COVID-19, male patients have almost three times the odds of requiring intensive treatment unit (ITU) admission (OR = 2.84; 95% CI = 2.06, 3.92) and higher odds of death (OR = 1.39; 95% CI = 1.31, 1.47) compared to females. With few exceptions, the sex bias observed in COVID-19 is a worldwide phenomenon. An appreciation of how sex is influencing COVID-19 outcomes will have important implications for clinical management and mitigation strategies for this disease.

[1] Centre for Adolescent Rheumatology Versus Arthritis at UCL, UCLH, GOSH, London, UK. [2] Centre for Rheumatology Research, Division of Medicine, UCL, London, UK. [3] Infection, Immunity and Inflammation Research and Teaching Department, UCL Great Ormond Street Institute of Child Health, London, UK. [4] NIHR Biomedical Research Centre at Great Ormond Street Hospital, London, UK. [5] Department of Paediatric Rheumatology, School of Child and Adolescent Health, Red Cross War Memorial Children's Hospital, University of Cape Town, Cape Town, South Africa. [6] The Francis Crick Institute, Crick African Network, London, UK. [7] These authors contributed equally: Elizabeth C. Rosser, Kate Webb, Claire T. Deakin. ✉email: kate.webb@uct.ac.za; c.deakin@ucl.ac.uk

                                                                                                                 1

A consistent feature of the ongoing coronavirus disease 2019 (COVID-19) pandemic, caused by the novel severe acute respiratory syndrome coronavirus 2 (SARS-CoV-2)[1], is the male bias towards severe disease[2–6]. Despite this, there are few analyses addressing whether this is a global rather than regional phenomenon. There are limited data that indicate whether the bias towards increased mortality in males is due to an increased proportion of infections in males, or a true representation of more severe disease. This gap in the literature highlights that sex remains an under-appreciated variable when interrogating outcomes in infectious diseases.

To address whether the reported sex-bias is validated by large-scale statistical analysis of global data, we have collected available case data from 90 reports including 46 different countries and 44 US states totalling 3,111,714 infected cases, and present a meta-analysis to investigate sex as a risk factor for SARS-CoV-2 infection, and COVID-19 morbidity and mortality. We demonstrate that while there is no difference in the proportion of males and females infected with SARS-CoV-2, males face higher odds of both intensive therapy unit (ITU) admission and death compared to females. The confirmation of this sex disparity with global data has important implications for the continuing public health response to this pandemic.

## Results

**Source selection and inclusion.** To interrogate the sex-bias in the global COVID-19 pandemic, 107 reports were gathered from across the world, from 1st January 2020 until the $1^{st}$ June 2020 (Fig. 1). Reports were included if they contained data on the total number of infections by sex, and the severity of disease as measured by admission to ITU and death ($n = 92$). Most reports ($n = 88$) were government websites, one report was from the British Medical Journal Global Health Blog website, and a minority comprised published research articles ($n = 3$), from which data were captured from full text. Importantly, sex disaggregated data for the United States of America (USA) were only available at individual state levels. Data from individual states were therefore analysed as separate regions to account for differences in how the data may have been collected. Reports were excluded if they did not report the overall number of infections by sex (one ITU case series and three mortality case series), or had small case numbers (less than five). Of the remaining 102 reports, multiple early reports originated from China, which were carefully examined for duplication. Following the exclusion of nine reports due to possible duplication, and one report from Canada with a large percentage of cases with unknown sex, a total of 92 reports from 47 different countries were analysed further (Supplementary Data 1)[7–98]. The 92 reports included three reports from China, the largest of which included data on confirmed cases by sex and mortality by sex, but not ITU admission by sex[9]. For this reason, the other two Chinese reports[10,11] were included in the analysis of ITU admission by sex, but were excluded from the analyses of confirmed cases and mortality by sex as it was likely these two reports overlapped with the larger report of Chinese cases[9].

Final reports and cases contributing to analysis: Ninety reports described the sex of 3,111,714 infected cases in 46 countries and 44 US states. Eight reports, representing 341,571 cases described 12,067 ITU admissions by sex. Seventy-one reports included mortality by sex, and after exclusion of the one possible overlapping report from China, 70 reports describing the sex of 2,751,115 COVID-19 cases and 214,361 related deaths were included in the final analysis. Counts of confirmed male cases and male deaths were calculated for 23 and 13 sources, respectively as these sources reported percentages and not counts. Thirty-four sources included a small proportion of cases of unknown sex and

17 sources included a small proportion of deaths of unknown sex (Supplementary Data 1).

Meta-analyses of infection, ITU and mortality risks: The proportion of male cases with COVID-19 in these reports was exactly half at 0.50 (95% confidence interval (CI) = 0.48,0.51; $p = 0.56$; $n = 3,111,714$), demonstrating that males and females have similar numbers of infections (Fig. 2). Male sex was associated with increased odds of ITU admission (odds ratio (OR) = 2.84; 95% CI = 2.06, 3.92; $p = 1.86 \times 10^{-10}$; $n = 341,571$) (Fig. 3), and increased odds of death (OR = 1.39; 95% CI = 1.31,1.47; $p = 5.00 \times 10^{-30}$; $n = 2,751,115$) (Fig. 4).

Funnel plots and sensitivity analyses indicated that the results for the estimated proportion of male cases and the estimated OR for ITU admission in men were unlikely to be influenced by reporting bias; however, the estimated OR for mortality in men may be an underestimate (Supplementary Information). Taken together these data highlight that, whilst there is no difference in the proportion of COVID-19 cases between sexes, men have a higher risk of ITU admission (OR 2.84) and death (OR 1.39).

## Discussion

Here we present a large-scale global statistical analysis to show that whilst males and females are at equivalent risk of infection, male sex is associated with the development of severe disease as measured by ITU admission (OR = 2.84; 95% CI = 2.06, 3.92; $p = 1.86 \times 10^{-10}$) and death (OR = 1.39; 95% CI = 1.31,1.47; $p = 5.00 \times 10^{-30}$). Despite this notable feature of the pandemic, sex is still not routinely reported in all available regional data.

Sex differences in the prevalence and outcomes of infectious diseases occur at all ages, with an overall higher burden of bacterial, viral, fungal and parasitic infections in human males[99–102]. Previous coronavirus outbreaks have demonstrated the same sex bias. The Hong Kong SARS-CoV-1 epidemic showed an age-adjusted relative mortality risk ratio of 1.62 (95% CI = 1.21, 2.16) for males[103]. During the same outbreak in Singapore, male sex was associated with an odds ratio of 3.10 (95% CI = 1.64, 5.87; $p \le 0.001$) for ITU admission or death[104]. The Saudi Arabian MERS outbreak in 2013 - 2014 exhibited a case fatality rate of 52% in men and 23% in women[105]. These data suggest that, whilst socio-economic factors may be influencing some aspects of the pandemic, fundamental differences in the immune response between males and females are likely to be a driving factor behind the significant sex-bias observed in the COVID-19 pandemic.

Sex differences in both the innate and adaptive immune system have been previously reported and may account for the female advantage in COVID-19. Within the adaptive immune system, females have higher numbers of CD4+ T cells[106–111], more robust CD8+ T cell cytotoxic activity[112], and increased B cell production of immunoglobulin compared to males[106,113]. Women report more severe local and systemic side effects, and produce higher antibody titres in response to the trivalent inactivated seasonal influenza vaccination (TIV)[100], as well as to most other pathogen vaccines[114]. More specifically, females achieve equivalent protective antibody titres to males at half the dose of TIV[115], with serum testosterone levels inversely correlating with TIV antibody titres[116]. Female B cells also produce more antigen-specific IgG in response to TIV[117]. These findings imply that females have an increased capacity to mount humoral immune responses compared to males, and together with the data from this meta-analysis, may have important implications for the development of vaccination strategies for COVID-19.

Females produce more type 1 interferon (IFN), a potent anti-viral cytokine, upon toll-like receptor 7 sensing of viral RNA than males[118–124], which is important for the early response in COVID-19[125]. Increased IFN production by females is associated

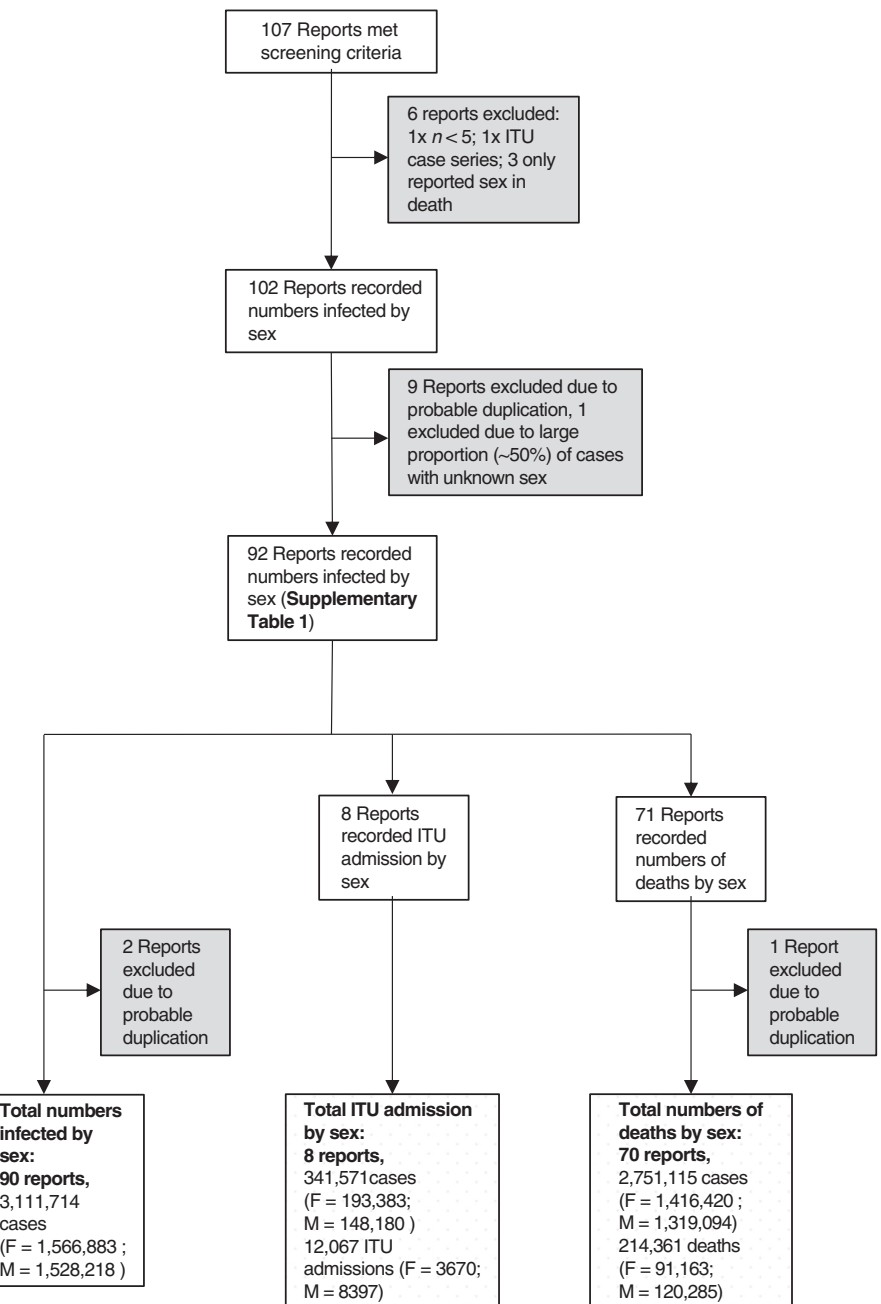

**Fig. 1 Study selection.** A total of 107 reports were found. Five reports were excluded as they did not report the total number of infections by sex (one intensive therapy unit (ITU) admission only case series and three mortality case series) and one report was excluded as it contained less than five cases. Of the 102 remaining reports, 9 were subsequently excluded due to possible duplication and 1 was excluded due to a large proportion of cases with unknown sex, yielding a total of 92 reports contributing to the analysis. These included three reports from China, which contributed differentially to the analysis. The largest report from China was used for analyses of confirmed cases and mortality by sex with the two other reports from China excluded from those analyses but used for the analysis of ITU admission by sex. This resulted in 90 reports of confirmed cases by sex, eight reports with data on ITU admission by sex and 70 reports with data on mortality by sex. Note that where totals for males and females do not add to the reported totals, this is because sex was unknown for some cases in the original source data.

with both sex hormone concentration and the number of X chromosomes present[119,121]. The X chromosome contains many immune-related genes[126], as evidenced by the existence of many X-linked immunodeficiency disorders[127]. Furthermore X-encoded immune genes may be variably expressed on both alleles in immune cells in females, increasing immune response diversity[128,129]. Oestradiol offers an advantage against infectious disease by augmenting T cell responses[112,130–132], increasing antibody production, somatic hyper-mutation and class

switching[133]. Oestradiol also increases abundance of neutrophils[134]; and monocyte/macrophage cytokine production[135].

An early report in pre-print suggested that the declining oestrogen levels in post-menopausal women may be associated with increased inflammatory cytokine production following infection with SARS-CoV-2[136]. This suggests a potential protective effect of oestradiol against the development of hyperinflammatory immune responses associated with mortality in COVID-19[137]. In contrast, the male sex hormone testosterone suppresses the

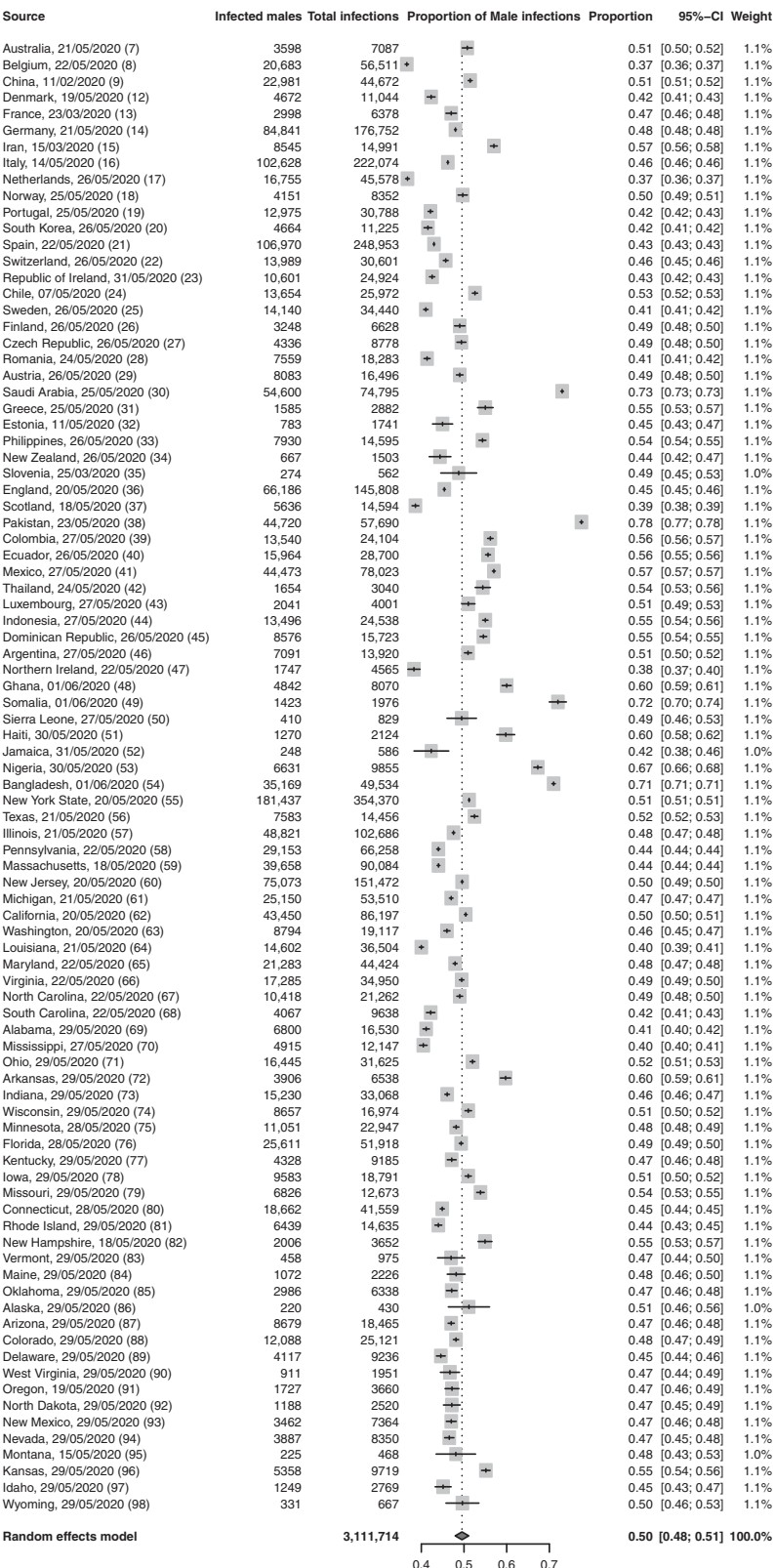

| Source | Infected males | Total infections | Proportion of Male infections | Proportion | 95%–CI | Weight |
|---|---|---|---|---|---|---|
| Australia, 21/05/2020 (7) | 3598 | 7087 | | 0.51 | [0.50; 0.52] | 1.1% |
| Belgium, 22/05/2020 (8) | 20,683 | 56,511 | | 0.37 | [0.36; 0.37] | 1.1% |
| China, 11/02/2020 (9) | 22,981 | 44,672 | | 0.51 | [0.51; 0.52] | 1.1% |
| Denmark, 19/05/2020 (12) | 4672 | 11,044 | | 0.42 | [0.41; 0.43] | 1.1% |
| France, 23/03/2020 (13) | 2998 | 6378 | | 0.47 | [0.46; 0.48] | 1.1% |
| Germany, 21/05/2020 (14) | 84,841 | 176,752 | | 0.48 | [0.48; 0.48] | 1.1% |
| Iran, 15/03/2020 (15) | 8545 | 14,991 | | 0.57 | [0.56; 0.58] | 1.1% |
| Italy, 14/05/2020 (16) | 102,628 | 222,074 | | 0.46 | [0.46; 0.46] | 1.1% |
| Netherlands, 26/05/2020 (17) | 16,755 | 45,578 | | 0.37 | [0.36; 0.37] | 1.1% |
| Norway, 25/05/2020 (18) | 4151 | 8352 | | 0.50 | [0.49; 0.51] | 1.1% |
| Portugal, 25/05/2020 (19) | 12,975 | 30,788 | | 0.42 | [0.42; 0.43] | 1.1% |
| South Korea, 26/05/2020 (20) | 4664 | 11,225 | | 0.42 | [0.41; 0.42] | 1.1% |
| Spain, 22/05/2020 (21) | 106,970 | 248,953 | | 0.43 | [0.43; 0.43] | 1.1% |
| Switzerland, 26/05/2020 (22) | 13,989 | 30,601 | | 0.46 | [0.45; 0.46] | 1.1% |
| Republic of Ireland, 31/05/2020 (23) | 10,601 | 24,924 | | 0.43 | [0.42; 0.43] | 1.1% |
| Chile, 07/05/2020 (24) | 13,654 | 25,972 | | 0.53 | [0.52; 0.53] | 1.1% |
| Sweden, 26/05/2020 (25) | 14,140 | 34,440 | | 0.41 | [0.41; 0.42] | 1.1% |
| Finland, 26/05/2020 (26) | 3248 | 6628 | | 0.49 | [0.48; 0.50] | 1.1% |
| Czech Republic, 26/05/2020 (27) | 4336 | 8778 | | 0.49 | [0.48; 0.50] | 1.1% |
| Romania, 24/05/2020 (28) | 7559 | 18,283 | | 0.41 | [0.41; 0.42] | 1.1% |
| Austria, 26/05/2020 (29) | 8083 | 16,496 | | 0.49 | [0.48; 0.50] | 1.1% |
| Saudi Arabia, 25/05/2020 (30) | 54,600 | 74,795 | | 0.73 | [0.73; 0.73] | 1.1% |
| Greece, 25/05/2020 (31) | 1585 | 2882 | | 0.55 | [0.53; 0.57] | 1.1% |
| Estonia, 11/05/2020 (32) | 783 | 1741 | | 0.45 | [0.43; 0.47] | 1.1% |
| Philippines, 26/05/2020 (33) | 7930 | 14,595 | | 0.54 | [0.54; 0.55] | 1.1% |
| New Zealand, 26/05/2020 (34) | 667 | 1503 | | 0.44 | [0.42; 0.47] | 1.1% |
| Slovenia, 25/03/2020 (35) | 274 | 562 | | 0.49 | [0.45; 0.53] | 1.0% |
| England, 20/05/2020 (36) | 66,186 | 145,808 | | 0.45 | [0.45; 0.46] | 1.1% |
| Scotland, 18/05/2020 (37) | 5636 | 14,594 | | 0.39 | [0.38; 0.39] | 1.1% |
| Pakistan, 23/05/2020 (38) | 44,720 | 57,690 | | 0.78 | [0.77; 0.78] | 1.1% |
| Colombia, 27/05/2020 (39) | 13,540 | 24,104 | | 0.56 | [0.56; 0.57] | 1.1% |
| Ecuador, 26/05/2020 (40) | 15,964 | 28,700 | | 0.56 | [0.55; 0.56] | 1.1% |
| Mexico, 27/05/2020 (41) | 44,473 | 78,023 | | 0.57 | [0.57; 0.57] | 1.1% |
| Thailand, 24/05/2020 (42) | 1654 | 3040 | | 0.54 | [0.53; 0.56] | 1.1% |
| Luxembourg, 27/05/2020 (43) | 2041 | 4001 | | 0.51 | [0.49; 0.53] | 1.1% |
| Indonesia, 27/05/2020 (44) | 13,496 | 24,538 | | 0.55 | [0.54; 0.56] | 1.1% |
| Dominican Republic, 26/05/2020 (45) | 8576 | 15,723 | | 0.55 | [0.54; 0.55] | 1.1% |
| Argentina, 27/05/2020 (46) | 7091 | 13,920 | | 0.51 | [0.50; 0.52] | 1.1% |
| Northern Ireland, 22/05/2020 (47) | 1747 | 4565 | | 0.38 | [0.37; 0.40] | 1.1% |
| Ghana, 01/06/2020 (48) | 4842 | 8070 | | 0.60 | [0.59; 0.61] | 1.1% |
| Somalia, 01/06/2020 (49) | 1423 | 1976 | | 0.72 | [0.70; 0.74] | 1.1% |
| Sierra Leone, 27/05/2020 (50) | 410 | 829 | | 0.49 | [0.46; 0.53] | 1.1% |
| Haiti, 30/05/2020 (51) | 1270 | 2124 | | 0.60 | [0.58; 0.62] | 1.1% |
| Jamaica, 31/05/2020 (52) | 248 | 586 | | 0.42 | [0.38; 0.46] | 1.0% |
| Nigeria, 30/05/2020 (53) | 6631 | 9855 | | 0.67 | [0.66; 0.68] | 1.1% |
| Bangladesh, 01/06/2020 (54) | 35,169 | 49,534 | | 0.71 | [0.71; 0.71] | 1.1% |
| New York State, 20/05/2020 (55) | 181,437 | 354,370 | | 0.51 | [0.51; 0.51] | 1.1% |
| Texas, 21/05/2020 (56) | 7583 | 14,456 | | 0.52 | [0.52; 0.53] | 1.1% |
| Illinois, 21/05/2020 (57) | 48,821 | 102,686 | | 0.48 | [0.47; 0.48] | 1.1% |
| Pennsylvania, 22/05/2020 (58) | 29,153 | 66,258 | | 0.44 | [0.44; 0.44] | 1.1% |
| Massachusetts, 18/05/2020 (59) | 39,658 | 90,084 | | 0.44 | [0.44; 0.44] | 1.1% |
| New Jersey, 20/05/2020 (60) | 75,073 | 151,472 | | 0.50 | [0.49; 0.50] | 1.1% |
| Michigan, 21/05/2020 (61) | 25,150 | 53,510 | | 0.47 | [0.47; 0.47] | 1.1% |
| California, 20/05/2020 (62) | 43,450 | 86,197 | | 0.50 | [0.50; 0.51] | 1.1% |
| Washington, 20/05/2020 (63) | 8794 | 19,117 | | 0.46 | [0.45; 0.47] | 1.1% |
| Louisiana, 21/05/2020 (64) | 14,602 | 36,504 | | 0.40 | [0.39; 0.41] | 1.1% |
| Maryland, 22/05/2020 (65) | 21,283 | 44,424 | | 0.48 | [0.47; 0.48] | 1.1% |
| Virginia, 22/05/2020 (66) | 17,285 | 34,950 | | 0.49 | [0.49; 0.50] | 1.1% |
| North Carolina, 22/05/2020 (67) | 10,418 | 21,262 | | 0.49 | [0.48; 0.50] | 1.1% |
| South Carolina, 22/05/2020 (68) | 4067 | 9638 | | 0.42 | [0.41; 0.43] | 1.1% |
| Alabama, 29/05/2020 (69) | 6800 | 16,530 | | 0.41 | [0.40; 0.42] | 1.1% |
| Mississippi, 27/05/2020 (70) | 4915 | 12,147 | | 0.40 | [0.40; 0.41] | 1.1% |
| Ohio, 29/05/2020 (71) | 16,445 | 31,625 | | 0.52 | [0.51; 0.53] | 1.1% |
| Arkansas, 29/05/2020 (72) | 3906 | 6538 | | 0.60 | [0.59; 0.61] | 1.1% |
| Indiana, 29/05/2020 (73) | 15,230 | 33,068 | | 0.46 | [0.46; 0.47] | 1.1% |
| Wisconsin, 29/05/2020 (74) | 8657 | 16,974 | | 0.51 | [0.50; 0.52] | 1.1% |
| Minnesota, 28/05/2020 (75) | 11,051 | 22,947 | | 0.48 | [0.48; 0.49] | 1.1% |
| Florida, 28/05/2020 (76) | 25,611 | 51,918 | | 0.49 | [0.49; 0.50] | 1.1% |
| Kentucky, 29/05/2020 (77) | 4328 | 9185 | | 0.47 | [0.46; 0.48] | 1.1% |
| Iowa, 29/05/2020 (78) | 9583 | 18,791 | | 0.51 | [0.50; 0.52] | 1.1% |
| Missouri, 29/05/2020 (79) | 6826 | 12,673 | | 0.54 | [0.53; 0.55] | 1.1% |
| Connecticut, 28/05/2020 (80) | 18,662 | 41,559 | | 0.45 | [0.44; 0.45] | 1.1% |
| Rhode Island, 29/05/2020 (81) | 6439 | 14,635 | | 0.44 | [0.43; 0.45] | 1.1% |
| New Hampshire, 18/05/2020 (82) | 2006 | 3652 | | 0.55 | [0.53; 0.57] | 1.1% |
| Vermont, 29/05/2020 (83) | 458 | 975 | | 0.47 | [0.44; 0.50] | 1.1% |
| Maine, 29/05/2020 (84) | 1072 | 2226 | | 0.48 | [0.46; 0.50] | 1.1% |
| Oklahoma, 29/05/2020 (85) | 2986 | 6338 | | 0.47 | [0.46; 0.48] | 1.1% |
| Alaska, 29/05/2020 (86) | 220 | 430 | | 0.51 | [0.46; 0.56] | 1.0% |
| Arizona, 29/05/2020 (87) | 8679 | 18,465 | | 0.47 | [0.46; 0.48] | 1.1% |
| Colorado, 29/05/2020 (88) | 12,088 | 25,121 | | 0.48 | [0.47; 0.49] | 1.1% |
| Delaware, 29/05/2020 (89) | 4117 | 9236 | | 0.45 | [0.44; 0.46] | 1.1% |
| West Virginia, 29/05/2020 (90) | 911 | 1951 | | 0.47 | [0.44; 0.49] | 1.1% |
| Oregon, 19/05/2020 (91) | 1727 | 3660 | | 0.47 | [0.46; 0.49] | 1.1% |
| North Dakota, 29/05/2020 (92) | 1188 | 2520 | | 0.47 | [0.45; 0.49] | 1.1% |
| New Mexico, 29/05/2020 (93) | 3462 | 7364 | | 0.47 | [0.46; 0.48] | 1.1% |
| Nevada, 29/05/2020 (94) | 3887 | 8350 | | 0.47 | [0.45; 0.48] | 1.1% |
| Montana, 15/05/2020 (95) | 225 | 468 | | 0.48 | [0.43; 0.53] | 1.0% |
| Kansas, 29/05/2020 (96) | 5358 | 9719 | | 0.55 | [0.54; 0.56] | 1.1% |
| Idaho, 29/05/2020 (97) | 1249 | 2769 | | 0.45 | [0.43; 0.47] | 1.1% |
| Wyoming, 29/05/2020 (98) | 331 | 667 | | 0.50 | [0.46; 0.53] | 1.1% |
| **Random effects model** | | 3,111,714 | | 0.50 | [0.48; 0.51] | 100.0% |

0.4    0.5    0.6    0.7

**Fig. 2 There is no observed sex difference in the proportion of people with COVID-19.** The table summarises the number of confirmed male COVID-19 cases and the total number of COVID-19 cases in 90 reports. The forest plot illustrates the estimated proportion of male cases for each report (grey boxes), with 95% confidence intervals (CI; horizontal black lines). The estimated pooled proportion of male cases (dark grey diamond) was 0.5 (95% CI = 0.48,0.51). A two-sided test confirmed the estimated pooled proportion was not significantly different from 0.5 ($p = 0.56$), indicating no difference between the proportions of male and female infections. Meta-analysis used a random effects model, which accounted for variance across reports and used the indicated weights for each report.

| Source | Males ITU | Males Total | Females ITU | Females Total | ITU odds ratio | OR | 95%–CI | Weight |
|---|---|---|---|---|---|---|---|---|
| China, 24/02/2020 (10) | 35 | 35 | 17 | 17 | | | | 0.0% |
| China, 07/02/2020 (11) | 22 | 75 | 14 | 63 | | 1.45 | [0.67; 3.15] | 8.3% |
| Denmark, 19/05/2020 (12) | 243 | 4672 | 91 | 6372 | | 3.79 | [2.97; 4.83] | 14.7% |
| France, 23/03/2020 (13) | 190 | 2998 | 95 | 3380 | | 2.34 | [1.82; 3.01] | 14.6% |
| Spain, 22/05/2020 (21) | 5358 | 106,970 | 2343 | 141,983 | | 3.14 | [2.99; 3.30] | 16.0% |
| Chile, 07/05/2020 (24) | 714 | 13,654 | 456 | 12,318 | | 1.44 | [1.27; 1.62] | 15.7% |
| Sweden, 26/05/2020 (25) | 1474 | 14,140 | 513 | 20,300 | | 4.49 | [4.05; 4.98] | 15.8% |
| Scotland, 18/05/2020 (37) | 361 | 5636 | 141 | 8950 | | 4.28 | [3.51; 5.21] | 15.1% |
| **Random effects model** | **148,180** | | **193,383** | | | 2.84 | [2.06; 3.92] | 100.0% |

0.2    0.5    1    2    5
Higher risk for women    Higher risk for men

**Fig. 3 Male sex is associated with a significantly increased risk of ITU admission within COVID-19 patients.** The table summarises the number of ITU admissions and total number of confirmed COVID-19 cases for each sex for $n = 8$ reports with complete data on SARS-CoV-2 infections and ITU admissions in males and females. The forest plot represents the estimated odds ratio (OR) for the association of ITU admission with male sex for each report (grey boxes), with 95% confidence intervals (CI; horizontal black lines). The estimated pooled OR (dark grey diamond) was 2.84 (95% CI = 2.06, 3.92). A two-sided test confirmed the estimated pooled OR was significantly different from 1 ($p = 1.86e^{-10}$). Meta-analysis used a random effects model with individual reports weighted using the indicated weights.

immune system: hypo-androgenism is associated with increased inflammatory cytokines, antibody titres, CD4/CD8 ratios, and natural killer cells, and a decrease in regulatory T cells[101,138]. Interestingly, testosterone-deprivation therapy for prostate cancer has been associated with improved outcomes for COVID-19, suggesting that suppression of the immune response by testosterone, as well as the protective effect of oestrogen, may underlie the observed sex-bias[139].

Age-related changes in the immune system are also different between sexes[140] and there is a marked association between morbidity/mortality and advanced age in COVID-19. For example, males show an age-related decline in B cells and a trend towards accelerated immune ageing[141,142]. This may further contribute to the sex bias seen in COVID-19.

Other biological factors may influence the sex-bias observed in this study. Expression of angiotensin converting enzyme 2 (ACE2) receptors – which facilitate SARS-CoV-2 viral entry and human to human transmission[143] – is different between the sexes[144,145]. Oestradiol may influence *ACE2* expression[146], and the gene for ACE2 is located in the X chromosome[147], which may render it susceptible to escaping X-inactivation in women.

Sex-based differences in co-morbidities that are associated with severe COVID-19 may also drive some of the differences observed in this study. However, due to the nature of these high level, publicly available summary data, metadata including age, ethnicity and comorbidities for individual cases are not available. The lack of adjustment for these factors limits our ability to accurately predict the role of sex in disease severity. Notably, there are no marked sex differences in the proportions of adults globally with hypertension (33% of women vs 36% of men)[148] or diabetes (9% of women vs 9.6% of men)[149], the most common reported comorbidities in hospitalised COVID-19 patients[150]. Once more data become available, future studies can adjust for additional factors using techniques such as mediation analysis.

Gender-based socio-cultural and behavioural differences could contribute to the sex difference seen in COVID-19 disease severity. Men are more likely to smoke[151], although smoking has not emerged as a clear risk factor for severe disease[144]. Men are less likely to wash their hands with soap after entering a restroom[152], and in many cultures, men may be more likely to leave the house and enter crowded areas. Unequal access to healthcare and testing between sexes may skew towards a male bias in infection rates. The data, however, show no difference in the numbers of infected cases between sexes overall, so gender differences in hygiene behaviours and testing are unlikely to explain the sex disparity in disease severity. Regional gender differences in health-seeking behaviours and access to care may predispose men towards access to hospital and ITU admission[153–156]. However, the ubiquitous nature of the sex-bias in these data argues for a true biological difference in the response to SARS-CoV-2 between sexes.

These large-scale data demonstrate that although there is no sex difference in the proportion of people infected with SARS-CoV-2, males are at a significantly higher risk of severe disease and death than females. Previous reports describe fundamental differences between sexes in the immune response to infection, which include a more robust antiviral innate interferon response and increased adaptive immunity towards viral antigens in females. In people infected with SARS-CoV-2 these differences are likely to lead to more effective viral control in females, which may contribute to the relatively lower risk of developing severe disease. Although further studies are needed, these data have implications for the clinical management of COVID-19 and highlight the importance of considering sex as a variable in fundamental and clinical research.

## Methods
**Search strategy**. An online search of government websites and published literature was performed by multiple researchers working remotely for regional data reports on COVID-19 cases that included sex as a variable from 01 January 2020 up until 01 June 2020 (Search terms: SARS-CoV-2/COVID-19/case/sex/country/data/death/ICU/ITU). In order to ensure unbiased representation from as many regions as possible, a cross check was done using the list of countries reporting data on 'Worldometer'[157], and an attempt was made to include as many regions reporting sex data as possible. Reports were translated using Google translate if they were not in English.

**Data selection, extraction and synthesis**. Reports were included if they contained sex as a variable in data describing case number, ITU admission, or mortality. Data were entered directly by individual researchers into an online structured data extraction table. For some sources, counts of male confirmed cases or male deaths were not provided, but percentages of male cases or male deaths were provided instead. To include these sources and avoid biases that might be introduced by their exclusion, we calculated counts of male confirmed cases and male deaths from the reported percentages with rounding to the nearest integer. We acknowledge that this approach assumes that the reported percentages are reflective of the true percentages. For some sources, data included confirmed cases and deaths of unknown sex. For these sources, the reported totals were used where the proportion of unknown sex was small (see Supplementary Data 1). This approach was preferred to excluding cases of unknown sex in order to avoid bias. The estimates represent the proportion of known male infections and ORs for mortality associated with known male sex, and will differ slightly from what the true values would be if the sex had been reported for all cases. Data were available at the level of country or regional summary data representing distinct individuals

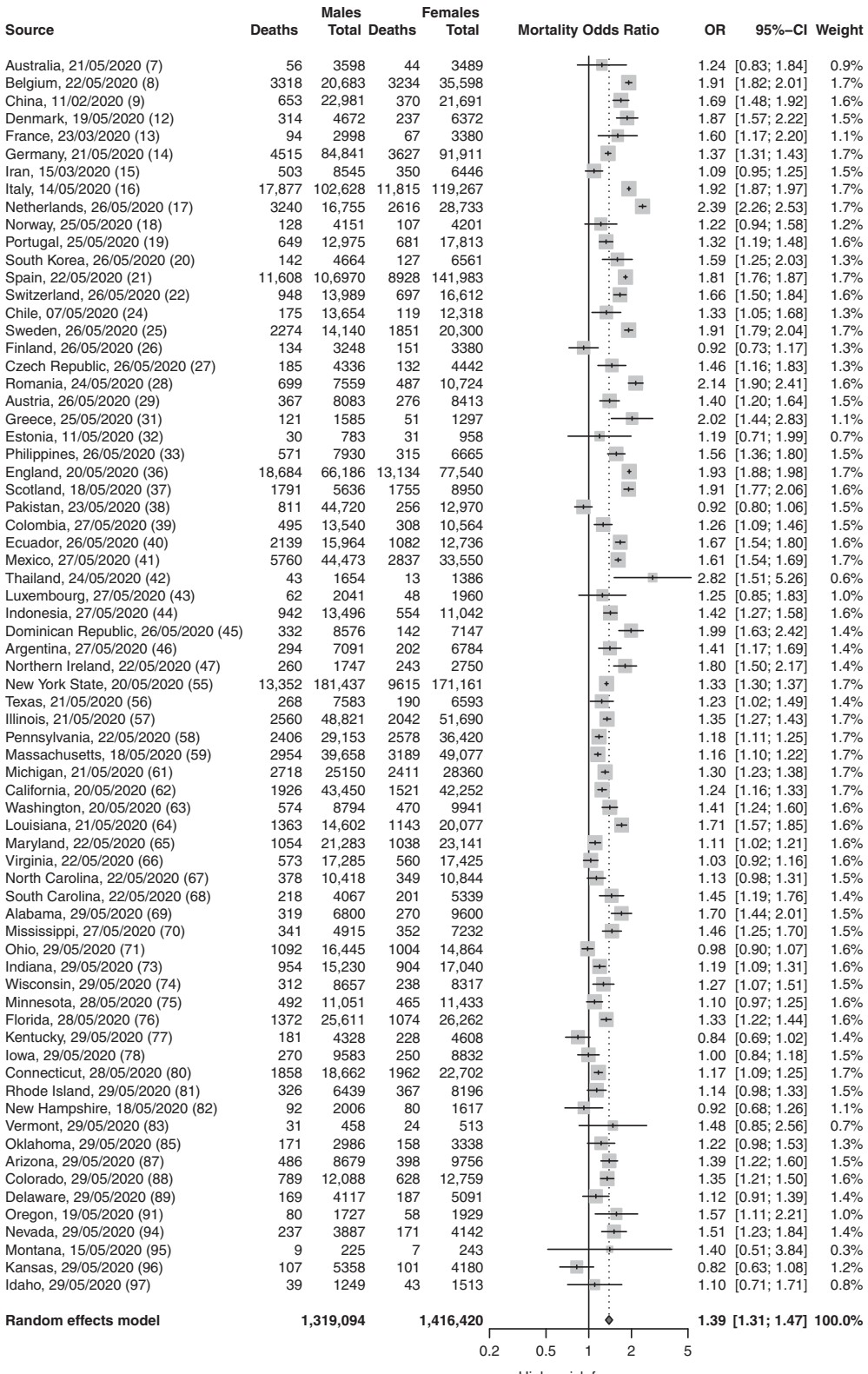

**Fig. 4 Male sex is associated with a significantly increased risk of mortality within COVID-19 patients.** The table summarises the number of deaths and total number of confirmed COVID-19 cases for each sex for $n = 70$ reports with complete data on infection and mortality in males and females. The forest plot illustrates the estimated odds ratio (OR) for the association of death with male sex for each report (grey boxes) with 95% confidence intervals (CI; horizontal black lines). The estimated pooled OR (dark grey diamond) was 1.39 (95% CI = 1.31, 1.47). A two-sided test confirmed the estimated pooled OR was significantly different from 1 ($p = 5.00e^{-30}$). Meta-analysis used a random effects model with individual reports weighted using the indicated weights.

for each report, but not at the level of covariates for all individuals within a study. Consequently, covariates such as lifestyle, comorbidities, testing method and case type (hospital vs. community) could not be controlled for. Data represent unique individuals at a single time snapshot. All data are provided in Supplementary Data 1 [7–98] and https://doi.org/10.25375/uct.12952151.

**Data analysis**. Meta-analysis was performed to estimate an overall proportion of male infected cases with 95% CI. This meta-analysis was two-sided and tested the null hypothesis that the proportion of male infected cases was 0.5. For this analysis, the classic inverse variance method for estimation of single proportions and standard errors was used, which uses logit-transformed proportions. The inverse variance method accounts for differing sample sizes of individual studies by weighting studies by the variance of their estimates, such that small studies with large variance have less weighting, and large studies with small variance have more weighting. A sensitivity analysis was also performed, in which the generalised linear mixed model (GLMM) method was used to estimate an overall proportion. This yielded identical results, indicating the differing assumptions of these different methods were inconsequential for these data.

Meta-analyses were also performed to estimate ORs with 95% CI associated with male sex for ITU admission and death, based on pooled average effect measures that were weighted according to the size and precision of each report. Random effects models were estimated and are reported rather than fixed effects models, since these do not assume uniformity across reports and account for variance between reports. The Mantel-Haenszel and DerSimonian-Laird methods were used to calculate the fixed effects and random effects estimates, respectively. Similar to the inverse variance weighting method, individual studies are weighted according to size and variance, and estimates were almost identical when the inverse variance weighting method was used. Both of these meta-analyses for the ITU admission and death outcomes were two-sided and tested the null hypothesis that the estimated OR was 1. Reports that did not contain the data required to calculate ORs were automatically excluded from meta-analyses.

Meta-analyses were performed using R version 3.6.1 and the "meta" package version 4.11-0 [158]. All R code and input data are available at https://github.com/claire-deakin/covid19-sex-bias and at https://doi.org/10.25375/uct.12952151.

**Reporting Summary**. Further information on research design is available in the Nature Research Reporting Summary linked to this article.

## Data availability
All data contributing to this analysis are provided in Supplementary Data 1 and published at figshare (https://doi.org/10.25375/uct.12952151). Screenshots of all sources from date of access are archived and may be available on request.

## Code availability.
The R code and input data used for all analyses are available at https://github.com/claire-deakin/covid19-sex-bias and at https://doi.org/10.25375/uct.12952151.

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

## Acknowledgements

KW is funded by the Crick African Network, African Career Accelerator Award (CANB0001/01). This work is funded by a Centre of Excellence (Centre for Adolescent Rheumatology Versus Arthritis) grant to LRW (21593) as well as grants from Medical Research Council (MR/R013926/1) and Great Ormond Street Children's Charity to LRW and NIHR Biomedical Research Centre at University College London Hospital to CC (BRC/III 525). ECR and NMdG are supported by a Medical Research Foundation Lupus Fellowship to ECR (MRF-057-0001-RG-ROSS-C0797). LRW and CD are supported by the NIHR Biomedical Research Centre at Great Ormond Street Hospital. HP is supported by a Versus Arthritis Studentship to CC (22203). The views expressed are those of the author(s) and not necessarily those of the NHS, the NIHR or the Department of Health. KW, ECR and CTD would like to thank this exceptional team for their dedication to getting these vital data gathered quickly and efficiently, with many working remotely, caring for young families and working on the frontline, and some working while unwell in quarantine. The funders of this study had no role in study design, data collection, data analysis, data interpretation, or writing of the report. The corresponding authors had full access to all the data in the study and had final responsibility for the decision to submit for publication.

## Author contributions

K.W., E.C.R., C.T.D., H.P., N.M.d.G., A.R., C.R. wrote the manuscript. C.T.D. analysed data, performed meta-analysis and generated figures. H.P., N.M.d.G., A.R., C.R., C.T.D. searched for, evaluated and gathered data. C.C. and L.R.W. edited and contributed to the manuscript. E.C.R., C.T.D., K.W., L.R.W. and C.C. obtained grant funding that supported this work.

## Competing interests

The authors declare no competing interests.
