## [Peer Review File · Nature Communications]

Editorial Note: Parts of this peer review file have been redacted as indicated to remove third-party material where no permission to publish could be obtained.

REVIEWER COMMENTS

Reviewer #1 (Sex/gender bias, viral immunity) (Remarks to the Author):

In this paper, the authors present the meta-analysis of a google-based online research on the gender-specific morbidity and mortality of Covid 19 disease induced by SARS coronavirus 2 (SARS-Co-2). The results show that despite the same infection rates in men and women, the rate of intensive care treatment and mortality is significantly higher in men than in women. In the following discussion, the authors shed light on the thematic background. On the one hand, they discuss gender-specific differences in previous outbreaks caused by other corona viruses such as SARS-CoV-1 and MERS but other infectious diseases. On the other hand, they discuss gender-specific biological influences on the immune system by chromosomal, hormonal and age-related factors.

The study underpins the previously observed trend for increased male vulnerability in the current Covid 19 pandemic. This could currently contribute to an increased attention of men with regard to taking specific measures for health prophylaxis and provides the basis for a strengthening of sex-specific basic research.

However, the study also reveals major weaknesses. On the one hand, it represents a snapshot in the midst of the current dynamic infection process, which could still change in the course of the pandemic. More serious, however, is the fact that socio-cultural aspects are not included in the evaluation and a critical analysis of the plain numbers is missing. For this reason, I believe that it is imperative to devote a part of the discussion to this study.

Furthermore, there are major weaknesses in the way the meta-analysis is carried out, which are discussed in detail below.

Major remarks:

1. The sources of information sought are not listed, they have only used Google-based analysis and have not used any databases. As a result, individual records are no longer available (citations 1), 11), 31)).

The authors did not mention whether a structured data abstraction form was used (minor), nor did they mention whether a funnel diagram or sensitivity analysis was performed (major), both of which are prerequisites for proper meta-analysis studies.

2. There is a particular need to discuss the cultural and socio-economic factors that could influence the higher vulnerability of men. This concerns in particular countries such as Iran and Saudi Arabia. The question here is whether, for example, women there have the same access to health systems as men.

Another example would be the male bias in infections with the MERS coronavirus (lane 146). This difference could be due to the fact that men are mainly responsible for the care of camelids.

An important complementation of the manuscript would be a research on further examples of infectious diseases with sex-specific differences in Western and non-Western countries listed in the study.

Otherwise, there is a risk of only obtaining simple plain numbers that might diminish or mislead the study results, which are in themselves very interesting.

3. The situation in the USA. There is no explanation here as to why, given the high mortality rates in the USA, no gender-specific analysis has been carried out e.g. in the New York State alone. In addition, there are anecdotal reports of increased morbidity and mortality rates among African Americans, which should be discussed and researched.

Minor:

5. A complete description of the pathogen (SARS-CoV-2) of Covid 19 should already be given in the abstract.

6. lane 91, supplementary list, Mortality numbers. From the text there should be 12 studies, in the supplementary list 13 studies are included.

7. lane 204, the influence of testosterone on vaccination should also be discussed here (see Furman et al. Systems analysis of sex differences reveals an immunosuppressive role for testosterone in the response to influenza vaccination, PNAS, 2014).

8. A short section on the evolutionary role of estrogens and testosterone would complement the article

9. For the discussion, the influence of pre-existing conditions on the course of Covid19 infection should be mentioned for the sake of completeness, as unrecognized pre-existing conditions occur more often in men, as men are less likely to seek medical advice compared to women.

Reviewer #2 (Hormone/immune crosstalk) (Remarks to the Author):

This study analysed for sex differences in the proportion of people infected with SARS-CoV-2, sex differences in ITU admission with COVID-19 and sex differences in mortality. The analysis included data from 29 reports involving 206,128 cases from multiple countries throughout the world (Europe, Middle East, Asia) so the numbers are robust. The statistical analysis seems appropriate and both fixed effects and random effects models were used. The results indicate a male predominance in ITU admissions and mortality in keeping with other coronavirus infections and sepsis in general. These data are of general interest to readers and, while the conclusions are not novel, this is the largest study describing the phenomenon of sex differences in COVID-19 disease outcomes.

Unfortunately, it is not a particularly well written paper. After a minimal methods and results of 2 and 1 paragraphs respectively, the authors then go on to review sex differences in immunity. This section is largely a repetition of what has already been published in the literature in far better reviews.

The results section is not adequate.

- The patient demographics should be described in results i.e. number of countries and regions represented in the data presented.
- Since different countries have different screening approaches it would be helpful to understand more about the screening approaches involved in the different studies included, i.e. were all participants symptomatic for COVID, were they hospital admissions or community-based, healthcare workers only? This is alluded to in the discussion which indicates that the authors have this information.
- They describe 29 studies as being included in the results section but in Fig. 1 it says 27 studies were included in the total number infected analysis. Why the discrepancy?
- There is no mention in the results that only 5 studies (42,454 cases) are included in the ITU series and 12 studies (171,104 case) in the mortality analysis. This must be made clear in methods and

results.

- Lines 124-6 33% and 36% sounds about equal so there is no ratio to reverse. If they claim these percentages are significantly different then the authors should provide the statistical analysis. Likewise, for diabetes where 9.6% and 9% are similar proportions.

The discussion is not very well structured and does not flow well.

- The authors jump from topic to topic in the same paragraph. For example, the second paragraph in discussion mixes sex and gender effects by first talking about society, behaviour and smoking (i.e. gender) then hypertension and diabetes propensity (sex factors) and then hand washing, leaving house and crowds (gender). It then discusses ACE2 expression in the same paragraph which is a separate issue and needs introduction.
- Line 170-185 – This section is muddled and appears more as a list than a systematic discussion. The authors start the sentence by discussing sex differences in “response to infection” but in the same sentence discuss “disease burden” rather than response. In the next sentence they mix infection response and infection rates. They do not qualify the reason more females suffer UTIs (due to anatomical differences in the male and female urogenital tracts).
- The authors discuss X-linked immunity and sex hormone effects on the immune system and then go on to discuss innate and adaptive immunity separately, having already discussed multiple sex differences in adaptive immunity in the former section. Furthermore, the authors contradict themselves saying testosterone decreases Tregs (line 238) and then males have more Tregs than females (line 262).

Minor comments

Line 43 “these data”

Line 129-30 “data however show no difference”

The figures are fine of the whole except one of the boxes in Fig 1 is missing a side.

Line 163 Please write MERS in full when first used.

Line 166 – Could a better verb than “collect” please be used?

Reviewer #3 (Systems immunology, sexual dimorphism) (Remarks to the Author):

The article conveys a short but effective point. Through looking at reports from different countries translated from different languages, the authors were able to amass high level summary data that showed that there is a higher percentage of male patients in the ICU than percentage of female patients, and a higher % of death for male patients. The other half of the article consists of literature search of similar findings indicating male may have more severe outcome in other infectious corona and influenza diseases. It's a short and very concise article and I don't have any issues in terms of methods and results.

However, in terms of novelty, there are already articles out there illustrating the same point.

<https://www.frontiersin.org/articles/10.3389/fpubh.2020.00152/full>

<https://jamanetwork.com/journals/jama/fullarticle/2765184>

<http://weekly.chinacdc.cn/en/article/id/e53946e2-c6c4-41e9-9a9b-fea8db1a8f51>

<https://www.medrxiv.org/content/10.1101/2020.04.24.20079046v1.full.pdf>

The novelty would lie in the fact that the data are from different countries.

Finally, the author did address that the data do not include age or co-morbidities and the consequence of such a lack of features. However, it would require a lot more research to effectively review whether COVID-19 is more severe for males or simply because of confounding variables. Therefore, this article

is much more of a supplementary report to trend we already know than a significant findings into the mechanistic difference in how the virus affects male vs. female.

minor: in-text citations should be after punctuation in this citation format.

Reviewer #4 (Bioinformatics, multivariate analysis) (Remarks to the Author):

This review paper tries to establish that while there is no difference in the infection rates of COVID-19 between sexes but there is a significant differences among the death rates of male vs. females among infected patients. Using a meta analysis, the authors demonstrated that male patients have more than double the odds of requiring intensive treatment unit admission and higher odds of death when compared to females. They also claim that females have more robust innate antiviral response and a better adaptive immune response to infection.

Specific Comments:

1. The way the authors performed the meta analysis is not very clear. Since various studies are being combined in a meta analysis framework, attention should be made on different study sizes and variance within and between studies. Consequently the meta analysis should be performed by taking care of the sample sizes and inverse of the variances. It is not clear how the authors are taking care of these issues.
2. It is not clear how the animal studies can be combined with human studies of different viral infections. Many cases they are not even comparable.
3. It is also not clear how 'females have more robust innate antiviral response and a better adaptive immune response to infection' are incorporated in the study.

Response to Reviewers

We would like to thank the reviewers for their time and consideration of our manuscript. We have now incorporated all the comments and amendments suggested below and substantially restructured the manuscript. A general theme was that whilst the manuscript was an important observation of large-scale data detailing the sex-bias in COVID-19 disease, it struggled as a review. With this in mind, we have now extensively reformatted the paper into a more traditional format. Another overarching comment from several reviewers was concerning the clarity of our methods and data handling. We found this feedback extremely constructive and by incorporating these comments we believe that it has significantly improved the transparency of our handling of the plain numbers and the subsequent analysis pipeline. We hope that the reviewers will now find our paper suitable for publication in Nature Communications.

Please see our response to the individual comments in italics below. We have used blue font for new text throughout the manuscript text. As the entire Supplementary section has been updated we have not used blue font here.

Referee #1

The article conveys a short but effective point. Through looking at reports from different countries translated from different languages, the authors were able to amass high level summary data that showed that there is a higher percentage of male patients in the ICU than percentage of female patients, and a higher % of death for male patients. The other half of the article consists of literature search of similar findings indicating male may have more severe outcome in other infectious corona and influenza diseases. It's a short and very concise article and I don't have any issues in terms of methods and results.

1. However, in terms of novelty, there are already articles out there illustrating the same point.

<https://www.frontiersin.org/articles/10.3389/fpubh.2020.00152/full>

<https://jamanetwork.com/journals/jama/fullarticle/2765184>

<http://weekly.chinacdc.cn/en/article/id/e53946e2-c6c4-41e9-9a9b-fea8db1a8f51>

<https://www.medrxiv.org/content/10.1101/2020.04.24.20079046v1.full.pdf>

The novelty would lie in the fact that the data are from different countries.

Thank you for this comment. We have now updated the text to reference these reports (citations 2-6 and 11) and highlight that this trend has been previously reported in smaller cohorts with our manuscript confirming the observations above in a more definitive manner with large scale data. We also highlight that the novelty of this manuscript is due

to the diversity of geographical regions represented and the large numbers reported. Importantly, this manuscript includes the largest analysis of US data, the world's most significant outbreak in terms of infections. In addition, we would also like to comment that only one of the papers above reports no difference in the proportion of infected cases between sexes, an important and novel feature of our study.

2. Finally, the author did address that the data do not include age or co-morbidities and the consequence of such a lack of features. However, it would require a lot more research to effectively review whether COVID-19 is more severe for males or simply because of confounding variables. Therefore, this article is much more of a supplementary report to trend we already know than a significant findings into the mechanistic difference in how the virus affects male vs. female.

We agree with the reviewer that our inability to build in covariates into our analysis such as age, co-morbidities and ethnicity is a significant limitation of the study. Due to the nature of the high-level data and wide-ranging regions that we have collected data from, unfortunately, this was impossible to incorporate into this analysis, from the studies analysed and their available data. We have now noted this in the methods (line 83, page 2) "Consequently, covariates such as lifestyle, comorbidities, testing method and case type (hospital vs. community) could not be controlled for" and also we have now discussed this limitation in the discussion (see line 241, page 5) of the revised manuscript document. Further studies from multiple regions are needed to integrate these co-variates with the sex-bias data reported here as well as more mechanistic-focused experimental work. This will form the basis of our future work and will be the subject of intense investigation by ourselves and we hope within the wider scientific community.

minor: in-text citations should be after punctuation in this citation format.

We would like to thank the reviewer for highlighting this error, this has been amended throughout.

Referee #2:

In this paper, the authors present the meta-analysis of a google-based online research on the gender-specific morbidity and mortality of Covid 19 disease induced by SARS coronavirus 2 (SARS-Co-2). The results show that despite the same infection rates in men and women, the rate of intensive care treatment and mortality is significantly higher in men than in women. In the following discussion, the authors shed light on the thematic background. On the one hand, they discuss gender-specific differences in previous outbreaks caused by other corona viruses such as SARS-CoV-1 and MERS but other

infectious diseases. On the other hand, they discuss gender-specific biological influences on the immune system by chromosomal, hormonal and age-related factors. The study underpins the previously observed trend for increased male vulnerability in the current Covid 19 pandemic. This could currently contribute to an increased attention of men with regard to taking specific measures for health prophylaxis and provides the basis for a strengthening of sex-specific basic research.

1. However, the study also reveals major weaknesses. On the one hand, it represents a snapshot in the midst of the current dynamic infection process, which could still change in the course of the pandemic.

We agree with this and therefore we have been continuously updating the data throughout the review period and have now included data up until 1st June 2020. We have now included 3,111,714 cases from 90 regions including 46 countries and 44 states of the USA. These updated data confirm this trend.

2. More serious, however, is the fact that socio-cultural aspects are not included in the evaluation and a critical analysis of the plain numbers is missing. For this reason, I believe that it is imperative to devote a part of the discussion to this study.

We have carefully considered this suggestion by the reviewer and agree that socio-cultural factors could have had a significant impact on the observations contained in this manuscript. However, as this trend is reported world-wide we strongly believe that biological factors are a driving force behind the observations presented in this manuscript. To highlight that we cannot discount socio-cultural factors, we have expanded the discussion to highlight that this is a confounder that should be taken into account when interpreting the data (please also see point 4 below):

Line 252, page 6: “Gender-based socio-cultural and behavioural differences could contribute to the sex difference seen in COVID-19 disease severity. Men are more likely to smoke,¹⁵² although smoking has not emerged as a clear risk factor for severe disease.¹⁴⁵ Men are less likely to wash their hands with soap after entering a restroom,¹⁵³ and in many cultures men may be more likely to leave the house and enter crowded areas. Unequal access to healthcare and testing between sexes may skew towards a male bias in infection rates. The data, however, show no difference in the numbers of infected cases between sexes overall, so gender difference in hygiene behaviours and testing are unlikely to explain the sex difference in disease severity. Regional gender differences in health seeking behaviours and access to care may bias men towards access to hospital and ITU admission.^{154–157} However, the ubiquitous nature of the sex

bias in these data argues for a true biological difference in the response to SARS-CoV2 between sexes.”

3. Furthermore, there are major weaknesses in the way the meta-analysis is carried out, which are discussed in detail below.

3.a. Major remarks:

- i. The sources of information sought are not listed, they have only used Google-based analysis and have not used any databases. As a result, individual records are no longer available (citations 1), 11), 31)).

Thank you for checking these. Due to the nature of continuously updated government reporting, many of the links cited will now no longer reflect the numbers collected. The date each source was accessed is stated in our figures and supplementary table 1. We also have screenshots of 84 of these from the time of data collection (not including the three sources published in journals), which we attach as a PDF, and seek the editor's discretion over inclusion of this within our supplementary materials. Regarding the three citations specified by the reviewer; citation 1 is omitted from the amended manuscript, but please see below screenshots of all three, with date visible. We suggest this may be a geographical access issue, and apologise for this.

Original Citation (1)

[Redacted]

Original Citation (11)

[Redacted]

Original Citation 31

[Redacted]

ii. The authors did not mention whether a structured data abstraction form was used (minor)

Thank you for this comment. We have now provided clarification in the text (in methods line 67, page 2), “Data were entered directly by individual researchers into an online structured data extraction table.”

iii)nor did they mention whether a funnel diagram or sensitivity analysis was performed (major), both of which are prerequisites for proper meta-analysis studies.

Thank you for this suggestion. We have now included a funnel diagram and sensitivity analysis (see Supplementary Figure 1) and added details regarding this to the methods (line 91, page 2) and supplementary sections.

4. There is a particular need to discuss the cultural and socio-economic factors that could influence the higher vulnerability of men. This concerns in particular countries such as Iran and Saudi Arabia. The question here is whether, for example, women there have the same access to health systems as men. Another example would be the male bias in infections with the MERS coronavirus (lane 146). This difference could be due to the fact that men are mainly responsible for the care of camelids. An important complementation of the manuscript would be a research on further examples of infectious diseases with sex-specific differences in Western and non-Western countries listed in the study. Otherwise, there is a risk of only obtaining simple plain numbers that might diminish or mislead the study results, which are in themselves very interesting.

We would like to thank the reviewer for this comment. Understanding the cultural and socio-economic factors that may influence the sex-bias in many different countries is a large and complex topic to address in this context given that many of the studies which met the criteria for this analysis did not contain this kind of information: therefore we agree that complementary studies should be used to explore this. If social-cultural factors were key driving forces to differences observed, one might expect to observe more discrepancies between countries in terms of infection and ITU intake. We believe that the fact that the sex-bias is observed in such a wide range of countries and continents suggests that biological factors are at least one driving force to these observations.

5. The situation in the USA. There is no explanation here as to why, given the high mortality rates in the USA, no gender-specific analysis has been carried out e.g. in the New York State alone.

We would like to thank the reviewer for this point. These data have been updated over the course of the review period and have now included data from the 44 states where data was available. Unfortunately, the USA does not publish sex-aggregated data as a whole.

6. In addition, there are anecdotal reports of increased morbidity and mortality rates among African Americans, which should be discussed and researched.

We agree that this is an important observation that requires in-depth analysis. Most of the studies and data sources used in this meta-analysis did not contain adequate data on ethnicity to allow us to perform an analysis on ethnicity. Therefore, while agreeing that this is a vital and highly important question, we hope that the reviewer agrees that as our manuscript focuses on sex-specific differences in COVID-19 outcomes that this analysis is beyond the scope of our study.

B. Minor:

1. A complete description of the pathogen (SARS-CoV-2) of Covid 19 should already be given in the abstract.

We have amended this as advised, lines 41-42, page 1 with citation #1.

6. line 91, supplementary list, Mortality numbers. From the text there should be 12 studies, in the supplementary list 13 studies are included.

We recognise this was inadequately clear and have edited the Results paragraph to better explain how we obtained the final numbers of studies that contributed to each analysis (lines 126-153 on pages 3-4, specifically). We have also edited the flow diagram in Figure 1 and the legend for Figure 1 to reflect this and to link better to Supplementary Table 1.

7. line 204, the influence of testosterone on vaccination should also be discussed here (see Furman et al. Systems analysis of sex differences reveals an immunosuppressive role for testosterone in the response to influenza vaccination, PNAS, 2014).

We would like to thank the reviewer for highlighting this important reference, this has now been added and discussed. Lines 195-201, pages 4-5: "More specifically, females achieve equivalent protective antibody titres to males at half the dose of TIV,¹¹⁷ with serum testosterone levels inversely correlating with TIV antibody titres.¹¹⁸ Female B cells also produce more antigen-specific IgG in response to TIV.¹¹⁹ This demonstrates that females have an increased capacity to mount humoral immune responses compared to males. These data may have important implications for the development of vaccination strategies for COVID-19."

8. A short section on the evolutionary role of estrogens and testosterone would complement the article.

Thank you for this suggestion- due to the comments of reviewer 3 below, we have amended the review to reflect more of a traditional discussion and had to shorten it

substantially and decrease its scope. Although this is an interesting point, we believe that this discussion would be outside the scope of the text in the current format.

9. For the discussion, the influence of pre-existing conditions on the course of Covid19 infection should be mentioned for the sake of completeness, as unrecognized pre-existing conditions occur more often in men, as men are less likely to seek medical advice compared to women.

We would like to thank the reviewer for this suggestion, we have made this clearer and this is included in the discussion:

Line 241-250, pages 5-6: “Sex-based differences in the certain co-morbidities that associate with severe COVID-19 may also drive some of the differences observed in this study. However, due to the nature of these high level, publicly available data, metadata including age, ethnicity and comorbidities for individual cases are not available. This limits our ability to accurately predict the role of sex in disease without adjusting for these factors. Notably, there are no marked sex differences in the proportions of adults globally with hypertension (33% of women vs 36% of men)¹⁴⁹ or diabetes (9% of women vs 9.6% of men),¹⁵⁰ the most common reported comorbidities in hospitalised COVID-19 patients.¹⁵¹ Once more data become available, future studies can adjust for additional factors using techniques such as mediation analysis.”

Referee #3:

This study analysed for sex differences in the proportion of people infected with SARS-CoV-2, sex differences in ITU admission with COVID-19 and sex differences in mortality. The analysis included data from 29 reports involving 206,128 cases from multiple countries throughout the world (Europe, Middle East, Asia) so the numbers are robust. The statistical analysis seems appropriate and both fixed effects and random effects models were used. The results indicate a male predominance in ITU admissions and mortality in keeping with other coronavirus infections and sepsis in general. These data are of general interest to readers and, while the conclusions are not novel, this is the largest study describing the phenomenon of sex differences in COVID-19 disease outcomes.

Unfortunately, it is not a particularly well written paper. After a minimal methods and results of 2 and 1 paragraphs respectively, the authors then go on to review sex differences in immunity. This section is largely a repetition of what has already been published in the literature in far better reviews.

We thank the reviewer for this comment and agree that it was ambitious to include a full review here. We have therefore shortened the review and included relevant points as a traditional discussion more focused on the meta-analysis rather than an extensive review. We hope that the reviewer will find the manuscript substantially improved.

A. The results section is not adequate.

1. The patient demographics should be described in results i.e. number of countries and regions represented in the data presented.

We have now updated the data within the manuscript to include 90 regions including 46 countries and 44 USA states. This has been clarified in the results section.

2. Since different countries have different screening approaches it would be helpful to understand more about the screening approaches involved in the different studies included, i.e. were all participants symptomatic for COVID, were they hospital admissions or community-based, healthcare workers only? This is alluded to in the discussion which indicates that the authors have this information.

Thank you for highlighting this important point. We have clarified in the methods (lines 80-85, page 2) to confirm that these data were not available:

“Data were available at the level of country or regional summary data representing distinct individuals for each report, but not at the level of covariates for all individuals within a study. Consequently, covariates such as lifestyle, comorbidities, testing method and case type (hospital vs. community) could not be controlled for. Data are available in Supplementary Table 1.”

3. They describe 29 studies as being included in the results section but in Fig. 1 it says 27 studies were included in the total number infected analysis. Why the discrepancy?

We recognise our description of final study numbers for each analysis was inadequately clear and have clarified this in the Results and Figure 1 using the numbers from the updated data series (see response to Reviewer 2, comment 6).

4. There is no mention in the results that only 5 studies (42,454 cases) are included in the ITU series and 12 studies (171,104 case) in the mortality analysis. This must be made clear in methods and results.

We would like to thank the reviewer for this comment. Please see the amended methods and results section, with the updated numbers and included a more in-depth explanation about total numbers. Figure 1 has also been updated accordingly.

5. Lines 124-6 33% and 36% sounds about equal so there is no ratio to reverse. If they claim these percentages are significantly different then the authors should provide the statistical analysis. Likewise, for diabetes where 9.6% and 9% are similar proportions.

Thank you for highlighting this, we were not trying to suggest that there were differences- we have shortened and simplified the section to:

Line 241-250, pages 5-6: “Sex-based differences in the certain co-morbidities that associate with severe COVID-19 may also drive some of the differences observed in this study. However, due to the nature of these high level, publicly available data, metadata including age, ethnicity and comorbidities for individual cases are not available. This limits our ability to accurately predict the role of sex in disease without adjusting for these factors. Notably, there are no marked sex differences in the proportions of adults globally with hypertension (33% of women vs 36% of men)¹⁴⁹ or diabetes (9% of women vs 9.6% of men),¹⁵⁰ the most common reported comorbidities in hospitalised COVID-19 patients.¹⁵¹ Once more data become available, future studies can adjust for additional factors using techniques such as mediation analysis.”

B. The discussion is not very well structured and does not flow well.

1. The authors jump from topic to topic in the same paragraph. For example, the second paragraph in discussion mixes sex and gender effects by first talking about society, behaviour and smoking (i.e. gender) then hypertension and diabetes propensity (sex factors) and then hand washing, leaving house and crowds (gender). It then discusses ACE2 expression in the same paragraph which is a separate issue and needs introduction.

We would like to thank the reviewer for this comment, as summarised above, based on the comments from this reviewer we have significantly re-formatted the manuscript to include a more traditional discussion instead of a meta-analysis. We hope that the reviewer will find the new version easier to follow and that the writing style has improved.

2. Line 170-185 – This section is muddled and appears more as a list than a systematic discussion. The authors start the sentence by discussing sex differences in “response to infection” but in the same sentence discuss “disease burden” rather than response. In the next sentence they mix infection response

and infection rates. They do not qualify the reason more females suffer UTIs (due to anatomical differences in the male and female urogenital tracts).

Thank you. As commented above, we have simplified and clarified this section as suggested. We hope that the referee will find the new format of the manuscript substantially improves its clarity.

3. The authors discuss X-linked immunity and sex hormone effects on the immune system and then go on to discuss innate and adaptive immunity separately, having already discussed multiple sex differences in adaptive immunity in the former section. Furthermore, the authors contradict themselves saying testosterone decreases Tregs (line 238) and then males have more Tregs than females (line 262).

Thank you. We hope that the reviewer finds our new version of the discussion clarifies these points.

C. Minor comments

1. Line 43 “these data”
We respectfully disagree as “data” are plural and therefore “these” is the correct demonstrative to use. We have not made this change.
2. Line 129-30 “data however show no difference”
Amended.
3. The figures are fine of the whole except one of the boxes in Fig 1 is missing a side.
Thank you- this figure has been redone.
4. Line 163 Please write MERS in full when first used.
Apologies and amended.
5. Line 166 – Could a better verb than “collect” please be used?
Changed to gather.

Reviewer #4

This review paper tries to establish that while there is no difference in the infection rates of COVID-19 between sexes but there is a significant differences among the death rates of male vs. females among infected patients. Using a meta analysis, the authors demonstrated that male patients have more than double the odds of requiring intensive treatment unit admission and higher odds of death when compared to females. They also claim that females have more robust innate antiviral response and a better adaptive immune response to infection.

Specific Comments:

1. The way the authors performed the meta analysis is not very clear. Since various studies are being combined in a meta analysis framework, attention should be made on different study sizes and variance within and between studies. Consequently the meta analysis should be performed by taking care of the sample sizes and inverse of the variances. It is not clear how the authors are taking care of these issues.

We have taken these comments into account, finding them extremely constructive in helping to increase the transparency of our data handling in the manuscript. The method section has been updated accordingly to include a better explanation of how differing study sizes and variances were accounted for and have added the following sentences:

Lines 92-95, page 2: "The inverse variance method accounts for differing sample sizes of individual studies by weighting studies by the variance of their estimates, such that small studies with large variance have less weighting, and large studies with small variance have more weighting."

Lines 104-110: "Random effects models were estimated and are reported rather than fixed effects models, since these do not assume uniformity across reports and account for variance between reports. The Mantel-Haenszel and DerSimonian-Laird methods were used to calculate the fixed effects and random effects estimates, respectively. Similar to the inverse variance weighting method, individual studies are weighted according to size and variance, and estimates were almost identical when the inverse variance weighting method was used."

2. It is not clear how the animal studies can be combined with human studies of different viral infections. Many cases they are not even comparable.

Thank you for this point, in the restructuring of the discussion we have taken out any reference to animal models.

3. It is also not clear how 'females have more robust innate antiviral response and a better adaptive immune response to infection' are incorporated in the study.

Thank you- this sentence has been deleted in the rewriting of the discussion. We hope that the reviewer finds the new version is clearer, we were referring to previously published studies rather than our own data.

REVIEWERS' COMMENTS

Reviewer #1 (Remarks to the Author):

1. The inclusion of the immensely large number of cases in the meta analysis has considerably improved the significance of the findings.
2. L 35 I would formulate this statement somewhat more cautiously, especially since there are conspicuous sex-specific differences in the expression of the human angiotensin converting enzyme 2 (ACE2), which are also addressed by the authors in the discussion and are probably due to genetic factors.
3. One of my main points of criticism regarding meta-analysis is now well addressed, the authors provide the missing information in the methods and supplements section.
4. Fig. 1, is there a mistake in the last row (weight), all studies are declared as 1.1%?
5. The inclusion of new immunological, COVID-19-specific data improves the discussion.

Prof. Dr. Hanna Lotter

Reviewer #2 (Remarks to the Author):

The new version of this paper now includes a meta-analysis of >3 million COVID-19 cases making it a more impressive and robust study. All my queries have been addressed and the new format of the paper is much better and more logical. There are just a few minor aspects of the discussion that are not written very well and would benefit from modification.

Lines 213-16 – the second half of this sentence does not flow from the first half and this sentence. It would be better to split into two sentences with the second one explaining X-inactivation properly.

Line 217 – “influencing T cells” in what way?

Line 234 – reference 115 is about IgG and not the “immune system” as the sentence suggests. Add some more relevant references please – you have many cited in the paper.

Reviewer #3 (Remarks to the Author):

Comments:

The manuscript is much improved with a more organized format and inclusion of more reports from around the world and reports from states in America into the meta-analysis. It shows that males are more likely to be admitted to ICU and have higher mortality than females. The main issue still resides in the fact that covariates such as lifestyle and co-morbidities, which cannot be accounted for in the innate nature of publicly available data. However, it does highlight the previously observed trend of males having more severe cases of COVID19 and could drive more attention toward sex differences that lead to such discrepancy.

Reviewer #4 (Remarks to the Author):

The authors have addressed all the criticisms raised by me. They have done a good job to take care of all the statistical issues.

My only concern about this paper is the data is always changing with time and the role of covariates getting more and more important.

REVIEWERS' COMMENTS

Reviewer #1 (Remarks to the Author):

1. The inclusion of the immensely large number of cases in the meta analysis has considerably improved the significance of the findings.

Thank you for your positive feedback.

2. L 35 I would formulate this statement somewhat more cautiously, especially since there are conspicuous sex-specific differences in the expression of the human angiotensin converting enzyme 2 (ACE2), which are also addressed by the authors in the discussion and are probably due to genetic factors.

This sentence has been deleted from the abstract: 'This suggests that fundamental differences in the immune response between males and females are likely to be the major driving factor underlying this sex-bias.'

3. One of my main points of criticism regarding meta-analysis is now well addressed, the authors provide the missing information in the methods and supplements section.

Thank you

4. Fig. 1, is there a mistake in the last row (weight), all studies are declared as 1.1%?

The weights are not actually all the same, but many of them appear to be close to 1.1% (there are a few that are 1.0%). The reasons why there are so many that are similar is (a) rounding of the weights to only 1 decimal place and (b) the large number of reports contributing to this analysis, which means that the weighting of individual reports is somewhat diluted. By contrast, the analysis represented in Figure 3 is derived from fewer reports with varying sample sizes. The magnitude of the weights are consistent with sample size but not linearly related. This ensures no individual reports are weighted too heavily such that weighting evens out with more and more reports.

5. The inclusion of new immunological, COVID-19-specific data improves the discussion.

Thank you

Prof. Dr. Hanna Lotter

Reviewer #2 (Remarks to the Author):

The new version of this paper now includes a meta-analysis of >3 million COVID-19 cases making it a more impressive and robust study. All my queries have been addressed and the new format of the paper is much better and more logical. There are just a few minor aspects of the discussion that are not written very well and would benefit from modification.

Thank you for your positive feedback.

Lines 213-16 – the second half of this sentence does not flow from the first half and this sentence. It would be better to split into two sentences with the second one explaining X-inactivation properly. Sentence split, 'The X chromosome contains many immune-related genes,¹²⁸ as evidenced by the existence of many X-linked immunodeficiency disorders¹²⁹; furthermore, X-encoded immune genes may be variably expressed on both alleles in immune cells in females, increasing immune response diversity'

We have split this sentence into two separate sentences as recommended.

Line 217 – “influencing T cells” in what way?

We have changed this to ‘augmenting T cell responses’

Line 234 – reference 115 is about IgG and not the “immune system” as the sentence suggests. Add some more relevant references please – you have many cited in the paper.

Apologies, this was an error. This has been corrected to the intended reference- Fulop T, Larbi A, Dupuis G, et al. Immunosenescence and inflamm-aging as two sides of the same coin: friends or foes? Front Immunol 8: 1960 (2017)

Reviewer #3 (Remarks to the Author):

Comments:

The manuscript is much improved with a more organized format and inclusion of more reports from around the world and reports from states in America into the meta-analysis. It shows that males are more likely to be admitted to ICU and have higher mortality than females. The main issue still resides in the fact that covariates such as lifestyle and co-morbidities, which cannot be accounted for in the innate nature of publicly available data. However, it does highlight the previously observed trend of males having more severe cases of COVID19 and could drive more attention toward sex differences that lead to such discrepancy.

Thank you for your positive feedback.

Reviewer #4 (Remarks to the Author):

The authors have addressed all the criticisms raised by me. They have done a good job to take care of all the statistical issues.

My only concern about this paper is the data is always changing with time and the role of covariates getting more and more important.

Thank you for your positive feedback.